
# Implementing Gas-to-Particle Partitioning of Semi-Volatile Inorganic Compounds in UCLALES-SALSA

Innocent Kudzotsa[1], Harri Kokkola[1], Juha Tonttila[1], Tomi Raatikainen[2], and Sami Romakkaniemi[1]

[1]Finnish Meteorological Institute, Atmospheric Research Center of Eastern Finland, P.O. Box 1627, 70211, Kuopio, Finland
[2]Finnish Meteorological Institute, P.O. Box 503, 00101, Helsinki, Finland

*Correspondence to:* Innocent Kudzotsa (innocent.kudzotsa@fmi.fi)

**Abstract.** We investigated the effect of inorganic semi-volatile compounds (SVC) $HNO_3$ and $NH_3$ on the cloud condensation nuclei (CCN) activity of aerosols and the subsequent changes in cloud properties. This was done by upgrading our state-of-the-art large eddy simulator - UCLALES-SALSA, which was modified to include the treatment of the condensation and dissolution of SVCs onto pre-existing aerosols and cloud droplets. The immediate effect of these SVCs on aerosols was to shift the aerosol

dry size distribution towards larger sizes as a result of their co-condensation with water vapor. Since the dry size of a CCN is the dominant factor determining its CCN activity, a marked increase in cloud droplet number concentration (similar to the Twomey effect) was noted both in zero- and three-dimensional simulations when gas-phase concentrations of SVCs were increased. As the overall amount of precipitation was small in the simulated stratocumulus case, the increase in droplet concentration led to a smaller mean size and reduced drizzle. Within clouds, the smaller droplets contain relatively higher amount of nitrate

than the larger ones, and as the drizzle is mainly formed through large droplets, the ammonium nitrate in-cloud scavenging is weaker than would be estimated based on average droplet composition. The model was also able to simulate the relatively more acidic interstitial particles than cloud droplets. However, below the cloud, condensation of gases on drizzling droplets quickly increases their overall wet scavenging efficiency compared to sulphate.

## 1 Introduction

The effects of aerosols on the Earth's energy and hydrological budgets have been a subject of research for a relatively long period of time; however, the anthropogenic effects of aerosols still impose uncertainties in the prediction of our future climate (Pörtner et al., 2019). Changes in aerosols loading and chemistry alter the Earth's energy balance by directly interacting with solar radiation (Charlson et al., 1992; Liao and Seinfeld, 1998; Myhre et al., 2009) and by altering the physical properties of clouds affecting their influence on radiation (Charlson et al., 1992; Haywood and Boucher, 2000; Lohmann and Feichter, 2005;

O'Donnell et al., 2011; Gettelman et al., 2012).

    A typical aerosol particle is comprised of water and multiple chemical compounds (Tao et al., 2017; Hodshire et al., 2019). These compounds include organic and inorganic compounds that can further be categorized as semi-volatile or non-volatile (Wehner et al., 2004). Under most atmospheric conditions, non-volatile compounds are normally found in the particle-phase of aerosols; while on the other hand, semi-volatile compounds can co-exist in all the three phases of matter depending on





the ambient conditions such as temperature and relative humidity (Kroll and Seinfeld, 2008). Hence, in real atmospheric conditions, semi-volatile compounds (SVC) are able to condense/evaporate onto/from pre-existing aerosol particles and can potentially modify aerosol size distributions, hygroscopicity and optical properties (Kokkola et al., 2003a; Topping et al., 2013). Whether or not a semi-volatile gas deposits or evaporates depends on several factors that include their volatility (Cousins et al., 1999) and also, on the type and amount of other soluble compounds constituted in the aerosol particles. Beyond compounds Henry's law constant, the volatility of SVCs depends on the degree of dissociation of the condensing gas (Saleh and Khlystov, 2009), which in turn is also dependent on other compounds in the solution (Seinfeld and Pandis, 2016).

While the Twomey effect (Twomey, 1974, 1977) and the lifetime indirect effect (Albrecht, 1989) that emanate from the increase in aerosol concentrations are no longer a conjecture, the presence of semi-volatile compounds (SVC) in the atmosphere has been suggested to cause similar effects on liquid clouds (Kulmala et al., 1995; Goodman et al., 2000; Kokkola et al., 2003b; Ma et al., 2010); although, their effects on cloud activation have not received much attention. Ammonia and nitrate are the dominant inorganic semi-volatile species in the troposphere (Adams et al., 1999, 2001). The main sources of ammonia are agricultural fertilizers, animal waste and biological decay (Dassios and Pandis, 1999) and can contribute up to two thirds of the total aerosol budget (Bouwman et al., 1997), while nitrates are chemically produced in the atmosphere from reactions involving ammonia and nitric acid. Nitric oxides, whose dominant sources are fossil fuel combustion, soils, biomass burning and lightning are the primary precursors of nitric acids (Bauer et al., 2007). The current typical concentrations of nitric acid range between 0.05 to 0.1 ppb in marine environments and about 1 ppb in continental boundary layer, although concentrations as high as 50 ppb in highly polluted environments are expectable (Kulmala et al., 1995; Bian et al., 2017b). The future concentrations of nitrate aerosols are estimated to increase in a short term due to predicted increase in its precursor gases, while sulphate aerosols are predicted to decrease (e.g., Pye et al., 2009; Xu and Penner, 2012; Hauglustaine et al., 2014; Paulot et al., 2016; Bauer et al., 2016).

It is therefore apparent that the present and future climatic importance of semi-volatile compounds is significant, and increasing number of global aerosol climate models is nowadays accounting them (Bauer et al., 2007; Bian et al., 2017a; An et al., 2019). The fraction of models studying nitrate has been low because of the complexity in modelling the partitioning process of $HNO_3$ and $NH_3$ between their gas and particle phases (Carslaw et al., 1994; Dassios and Pandis, 1999). What has been a major drawback in studying the effects of nitrates and ammonia on climate is that most of the numerical techniques available for solving the dissolution equations and the accompanying equilibrium models required to calculate the liquid-phase thermodynamics are computationally expensive (Bauer et al., 2007). As a result there are currently no studies that have assessed and quantified how effectively the semi-volatile compounds are sequestered from the atmosphere by clouds and precipitation because most available cloud activation parameterizations do not ingest ammonia and nitrate compounds condensing during activation process (Romakkaniemi et al., 2005a). Most studies that have been conducted so far have used single column or cloud parcel models (Kulmala et al., 1995; Romakkaniemi et al., 2005b; Topping et al., 2013; Crooks et al., 2016) and they have all agreed that the presence of trace semi-volatile gases modify the size distributions and the hygroscopic properties of aerosols (Bowman and Ben-Akiva, 2001; Kokkola et al., 2003a; Romakkaniemi et al., 2005b; Huffman et al., 2009; Cappa and Jimenez, 2010) and hence, the propensity of those solute aerosols to nucleate a cloud droplet and their interaction with radia-



tion (Ma et al., 2010; Makkonen et al., 2012; Lowe et al., 2018). Calculating the lifetime is complicated because it depends on the RH and temperature if these compounds are in the gas or aerosol phase, and consequently it is not straightforward to use the same cloud activation parameterizations as for non volatile aerosol compounds, which simply assume the same aerosol dry size and composition distribution within and below the cloud.

In order to assess for the first time, the efficiency with which these semi-volatile gases are sequestered from the atmosphere, we extended the Sectional Aerosol module for Large Scale Applications (SALSA) to include the treatment of these gases. This was achieved by implementing the dissolution of nitrate and ammonia into the SALSA module of our state-of-the-art Large Eddy Simulator (LES), UCLALES-SALSA. The part of the atmosphere pertinent to this work is the lower troposphere where liquid clouds exist. In the following section, we briefly describe the model and the scheme that treats the deposition

and evaporation of these semi-volatile gases and its implementation in UCLALES-SALSA. In Section 3, we compare the performance of this new scheme in UCLALES-SALSA against the performance of a more detailed cloud parcel model (CPM). Finally, the analysis of a three-dimensional simulation involving idealized gas-phase scenarios of nitrate and ammonia are shown and discussed in Section. 4.3, while conclusions are given in the last section of the paper.

## 2   The Models

In order to conduct this study, the condensation routine of the SALSA module (Tonttila et al., 2017) was improved and upgraded to include the dissolution of semi-volatile gases ($HNO_3$ and $NH_3$) on aerosols, clouds and precipitation species. The SALSA module, originally developed by Kokkola et al. (2008) is coupled to the Large Eddy Simulator (UCLALES) (Stevens et al., 2005) to give a state-of-the-art model, UCLALES-SALSA (Tonttila et al., 2017). This inventive coupling between a well-characterized LES and a detailed aerosol-cloud microphysics model enables the complex interplay between aerosols and cloud

to be accurately disentangled. UCLALES-SALSA has already been used successfully in past studies, e.g. by Tonttila et al. (2017) to study how the microphysical interactions between aerosols and clouds affect the boundary layer dynamics and by Boutle et al. (2018) to study the aerosol-fog interaction and its transition to well-mixed radiation fog. Another more detailed model, a Cloud Parcel Model (CPM) (Kokkola, 2003) was used as the reference model for the purposes of evaluating the performance of the upgraded version of SALSA. UCLALES-SALSA is originally written in FORTRAN-90 although the code

is gradually migrating towards FORTRAN-2013. For parallel computing, the Message Passing Interface (MPI) library is used and the parallelization strategy is based on two-dimensional horizontal blocking of the model domain, while the output is written out in Network Common Data Form (NetCDF) format.

### 2.1   An Overview of SALSA

SALSA was first developed as a sectional aerosol module built for large-scale models such as climate modelling which require

solving physical and chemical processes with accuracy and computational efficiency, simultaneously (Kokkola et al., 2008). In order to meet these criteria, the module design was based on three fundamental aspects: firstly, the resolution of the aerosol size distribution should be dependent on the relevance and importance of the aerosol processes or of the region to the problem





under investigation. Secondly, the different chemical compounds of aerosols are concentrated on different parts of the aerosol size spectrum and finally, the importance of different microphysical processes varies strongly across the aerosol size spectrum.

This SALSA version of Kokkola et al. (2008) has since been modified into an extended version of SALSA that includes cloud and precipitation processes (Tonttila et al., 2017; Kokkola et al., 2018), which is the version used in this study. This version is
a two-moment, sectional aerosol-cloud model that has the number and mass concentrations of aerosol, cloud and precipitation particles as prognostic variables. The key microphysical processes treated in this model include coagulation, condensation, cloud activation and auto-conversion of cloud droplets into precipitation. Although not included in this work, the ice-phase has also been incorporated into SALSA (Ahola et al., 2020). In these simulations, a total of ten size sections has been used for aerosols and seven size bins have been applied for cloud droplets, while precipitation has 20 volume doubling size sections.

## 2.2   The Clouds and Precipitation

The interaction and growth of particles by coagulation between the various species of hydrometeors and aerosols are resolved using the semi-implicit approach of Jacobson (2005) and a detailed description of how it is modelled is provided in Tonttila et al. (2017). In UCLALES-SALSA, the saturation ratio is explicitly predicted by taking into account both the radiative and dynamic effects influencing the thermodynamic properties of an air parcel. As a result, for cloud droplet activation, the wet
diameters of aerosol particles are resolved first; and when the diameters exceed the critical diameter corresponding to the resolved supersaturation ratio then those aerosols are activated into cloud droplets. This approach enables cloud activation at any part of the cloud profile unlike most techniques that only allow activation at cloud-base (Rogers and Yau, 2006; Hoffmann, 2017). Most importantly, this approach makes UCLALES-SALSA ideal for simulating more complex situations such as fog prediction where droplet activation occurs mainly at the fog-top and within the fog layer owing respectively to radiative cooling
and turbulence, which are the main drivers of water saturation ratio in fogs (Maalick et al., 2016; Boutle et al., 2018).

Because of the complexity and ambiguity associated with the definition and formation of precipitation from cloud droplets, two auto-conversion techniques have been developed for SALSA. The first one is a commonly used auto-conversion technique implemented for the formation of drizzle. This technique assumes that within every cloud bin, cloud droplets are characterized by a lognormal distribution whose geometric mean diameter corresponds to the mean diameter of that bin. Therefore, by
prescribing a standard deviation and a critical threshold for drizzle diameter, $d_0$, the mass concentration of cloud droplets transferable to the first bin of precipitation corresponds to the integral mass of particles larger than $d_0$ on the assumed lognormal distribution. The transferred number concentration of drizzle droplets is derived from the transferred mass concentrations and the initial size, $d_0$ prescribed for the smallest precipitation bin. The second technique is the one used in these simulations and is based on the self-coagulation of cloud droplets and cross-coagulation of aerosol particles and cloud droplets. When the size
of droplet formed in coagulation exceeds the critical threshold, which in this case is 20 µm, then the particle is transferred in the corresponding precipitation bin. This approach allows realistic representation of the whole cloud droplet size distribution.





## 2.3 Condensation, Dissolution and Evaporation

In order to conduct this assessment, the condensation process of UCLALES-SALSA was improved and further upgraded to include the dissolution of $HNO_3$ and $NH_3$. Thus, the condensation of water vapour onto the different size sections of aerosol and hydrometeor particles is calculated by solving the condensation and dissolution equations semi-implicitly at every time step in every grid point. This is done by a novel approach of combining the Analytical Predictor of Condensation (APC) and the Analytical Predictor of Dissolution (APD) schemes of Jacobson (2005). These two schemes are computationally efficient because of their non-iterative nature and they are also numerically stable and mass-conserving, which makes them computationally efficient and handy for models designed for large-scale applications.

Dissolutional and condensational growth are non-equilibrium time-dependent processes and they sometimes happen much faster (especially onto smaller particles) than the current timestep of 1 s being used in the model, which is chosen as a compromise between accuracy and computational time. As a result, the APC scheme exhibits an oscillatory behavior in the smaller size sections of the particle spectrum, especially in sub-saturated environments when particles are growing by condensation. This oscillatory behavior had been circumvented in the previous version of UCLALES-SALSA by calculating the wet sizes of particles as an equilibrium solution based on the molalities of different particle species when RH was below 98 % (Tonttila et al., 2017). But this technique is not computationally efficient, and hence, in this new version, we avoid this oscillatory behavior by treating the diffusion of water vapour onto particle surfaces in sub-saturated environments (i.e RH < 100 %) using the APD scheme, which in fact, produced quite stable, accurate and non-oscillatory results, although the APD is not designed for water. When comparing SALSA with the CPM, the APD scheme proved to be well suited for calculating the condensation of water in subsaturated conditions, while in-cloud, it failed to reproduce the sudden gradients in concentrations. Therefore, the APC scheme is currently only being used for condensation when RH > 99.9 %. As for evaporation of water, the APC scheme is used in all conditions.

Secondly, as mentioned above, condensational and dissolutional processes may occur much faster than the model time step especially for smaller particles. Therefore, in order to increase the temporal resolution with which these episodes are captured in UCLALES-SALSA, selective sub-timestepping was applied specifically for both the APC and the APD schemes in order to ensure that all the particle sizes are accurately equilibrated. The sub-timesteps are staggered by an order of magnitude, so that the shorter the sub-timesteps are, the more accurate the scheme becomes; however, we noted that beyond five substeps, the incentive for substepping drastically diminishes while the computational expense increases substantially and as a result, we limited the substeps to five. The implementation of the APC scheme is described below, while the APD scheme is described in Subsection. 2.3.2.

### 2.3.1 The Analytical Predictor of Condensation (APC)

The Analytical Predictor of Condensation (APC) is one of the several techniques available to implicitly solve the non-equilibrium time-dependent condensation and evaporation processes (Jacobson, 2005). Although the basic equations are the same as those presented in Jacobson, we also shortly present them here.





For the APC technique, the rate of change of particle phase concentration, $\frac{dc_{q,i,t}}{dt}$ of condensing gas $q$ in size section $i$ is defined as:

$$\frac{dc_{q,i,t}}{dt} = k_{q,i,t-h}(C_{q,t} - S'_{q,i,t-h}C_{q,s,i,t-h}) \tag{1}$$

where, $k_{q,i,t-h}$ is known as the mass transfer coefficient (see Jacobson (2005) for details), $h$ is the model time-step, while $C_{q,t}$

is the gas-phase concentration of the condensing gas in the current time-step. $S'_{q,i,t-h}$ is the equilibrium saturation ratio and $C_{q,s,i,t-h}$ is the uncorrected saturation vapour mole concentration of the condensing gas. By integrating Eqn. 1 for the final aerosol concentration, a non-iterative solution to the growth equation can be determined as follows.

$$c_{q,i,t} = c_{q,i,t-h} + hk_{q,i,t-h}(C_{q,t} - C_{q,s,i,t-h}) \tag{2}$$

It is apparent that Eqn 2 cannot be immediately solved analytically at the current timestep since $C_{q,t}$ is currently unknown.

Therefore, $C_{q,t}$ is resolved first by assuming that the sum of particle-phase and gas-phase concentrations is conserved between the previous and the current time steps to get Eqn. 3, where all terms on the right hand side of the equation are resolvable from the previous timestep and $C_{q,t}$ can then be used in equation 2:

$$C_{q,t} = \frac{C_{q,t-h} + h\sum\limits_{i=1}^{N_B}(k_{q,i,t-h}S'_{q,i,t-h}C_{q,s,i,t-h})}{1 + h\sum\limits_{i=1}^{N_B}k_{q,i,t-h}} \tag{3}$$

### 2.3.2    The Analytical Predictor of Dissolution (APD)

The condensational (Eqn. 1) and dissolutional growth equations are technically the same, except that the saturation vapour pressure in condensation can be calculated from the Clausius Clapeyron equation, whereas in dissolution, the saturation vapour pressure of the condensing gas is dependent on the molality of that gas already in the particle phase and is governed by the Henry's law. Hence, the non-iterative solution for the rate of change of particle phase concentration, $\frac{dc_{q,i,t}}{dt}$ during dissolution is:

$$c_{q,i,t} = \frac{H'_{q,i,t-h}C_{q,t}}{S'_{q,i,t-h}} + \left(c_{q,i,t-h} - \frac{H'_{q,i,t-h}C_{q,t}}{S'_{q,i,t-h}}\right) \times \tag{4}$$

$$exp\left(-\frac{hS'_{q,i,t-h}k_{q,i,t-h}}{H'_{q,i,t-h}}\right) \tag{5}$$

where, $H'_{q,i,t-h}$ is the effective Henry's law coefficient for the condensing gas, which again cannot be resolved directly because $C_{q,t}$ is currently unknown. Therefore by applying mass conservation for the condensing species between the current and the previous time steps,

$$C_{q,t} = \frac{C_{q,t-h} + \sum\limits_{i=1}^{N_B}\left\{c_{q,i,t-h}\left[1 - exp\left(-\frac{hS'_{q,i,t-h}k_{q,i,t-h}}{H'_{q,i,t-h}}\right)\right]\right\}}{1 + \sum\limits_{i=1}^{N_B}\left\{\frac{H'_{q,i,t-h}}{S'_{q,i,t-h}}\left[1 - exp\left(-\frac{hS'_{q,i,t-h}k_{q,i,t-h}}{H'_{q,i,t-h}}\right)\right]\right\}} \tag{6}$$





which can be substituted back into Eqn. 4. The full development of these two schemes is found in Jacobson (2005)

The Henry's coefficient is an important parameter that governs the partitioning between the gas and the particle phase concentrations of the condensable gas (Kuosa et al., 2004). It depends on the degree of dissociation of the dissolved gas, which is also affected by the degree of dissociation of other dissolved species and most importantly, on the equilibrium saturation

vapour pressure of the condensing gas (Jacobson et al., 1996). Therefore, in order to resolve this complex thermodynamic interplay within the aerosol particle there are several thermodynamic schemes of varying complexity (Zhang et al., 2000) that have been developed specifically to treat this liquid-phase thermodynamics problem e.g. EQUIL which was later on upgraded to KEQUIL of Bassett and Seinfeld (1983, 1984), ISORROPIA of Nenes et al. (1998); Fountoukis and Nenes (2007), AIM of Clegg et al. (1998) and PD-FiTE of Topping et al. (2005a, b) and several others. In this work, we used the AIM model primarily

for the validation of the model during one dimensional simulations. However, for the real three dimensional simulations using the LES, we switched onto ISORROPIA because it is computationally much faster than AIM.

The speed of thermodynamic calculation is relevant as the thermodynamics is called twice within every substep. Once in the beginning, and for the second time after the condensation of water and nitrate is calculated. This is because the efficient Henry's law coefficient of both compounds is highly dependent on the acidity of the aerosol, and adding the second call was

found to remarkably improve the accuracy of the solution when compared to the reference model. In addition, to ensure that there are no unwanted oscillations in the aerosol phase concentration, even for smallest particle sizes, a limiter was also added to limit the maximum change in concentration within a given substep. The limiter meant that the change in particle-phase concentration in the current timestep does not exceed 95 % of the total concentration in the previous timestep. We assume all aerosols to remain as liquid solutions in these simulations.

## 20 2.4 The Cloud Parcel Model (CPM)

To evaluate, how well the improved SALSA simulates the partitioning of $HNO_3$ and $NH_3$, we compare it to a cloud parcel model which simulates the partitioning with high precision. The cloud parcel model used in this study is a detailed sectional aerosol-cloud model that solves the ordinary differential equations for several microphysical and dynamical processes including the non-equilibrium growth equations for a population of aerosols in a rising or a stationary air parcel, liquid-phase chemical

thermodynamics and adiabatic equations. This is done using the double precision Livermore Solver for Ordinary Differential Equations (DLSODE), which is a FORTRAN solver for ordinary differential equations. For the liquid-phase thermodynamics, the CPM uses the AIM thermodynamic model (Clegg et al., 1998). For the full description of this cloud parcel model, the reader is referred to Kokkola (2003).

## 3 Model Evaluation

The evaluation of the model was performed by comparing the predictions of SALSA against those of the cloud parcel model. This is because there are no coincident measurements available, which the model could be compared against; and moreover, the CPM is a well-tested model that has already been used in several previous studies: for instance, in similar studies by (e.g.,





**Table 1.** The distribution parameters of the bimodal lognormal distribution assumed in both SALSA and the CPM models.

| Parameter | $1^{st}$ mode | $2^{nd}$ mode |
|---|---|---|
| Number concentration, $cm^{-3}$ | 200 | 20 |
| Geometric mean, μm | 0.7 | 0.7 |
| Standard deviation ($\sigma$) | 1.35 | 1.35 |

Kokkola, 2003; Romakkaniemi et al., 2005a, 2009, 2011). Since the only major relevant changes performed to the UCLALES-SALSA version of (Tonttila et al., 2017) were related to the condensation process and the addition of the dissolution routine, we shall focus our evaluation on these two processes.

### 3.1 Condensation and Evaporation of Water Vapor

In order to evaluate the condensation process in SALSA, we performed an ensemble of model runs in which the aerosol and environmental conditions were set as follows. For aerosols, a bi-modal lognormal distribution was assumed for the particle phase sulphate aerosols and the corresponding parameters for the aerosol distribution are given in Table 1. Condensation was simulated for an air parcel in consecutive ascending and descending motions and thus subject to adiabatic cooling and warming. The ascend was started from 1000 hPa pressure level and continued until saturation and into the cloud, followed by a descend

back to the initial pressure level. Therefore, adiabatic ascent and descent are the only factors driving the temperature changes of this air parcel. The same aerosol specifications and cooling rates were applied in both the CPM and SALSA.

The perturbed parameter ensemble approach was employed here; this involved 100 different model runs in which the parcel's initial temperature and ascent rate were simultaneously perturbed randomly. The parcel's initial temperature ranged between 268 and 278 K, while the parcel's ascent rate ranged between 0.005 and 2 $ms^{-2}$. These are typical ranges of updraft speeds

observed or predicted in real clouds and cloud models ranging from stratocumulus to cumulus type clouds (e.g, Sheffield et al., 2015; Saleeby et al., 2016). On the other hand, the initial relative humidity and pressure were kept constant at 98 % and 1000 hPa, respectively. Although the models were both able to also treat the condensation of SVCs in addition to water vapour, these other condensible gases were not included in these ensemble simulations. For the sake of easy reference in latter sections, we shall designate this model configuration or test as Ensemble A.

Fig. 1a shows the ensemble mean of how the aerosol size distributions evolved with time between the two models. The mean size distribution from SALSA (blue curve) is superimposed upon the mean size distribution from the CPM (red curve). As time evolves, while the air parcel rises adiabatically, the temperature of the air parcel drops raising the parcel's saturation ratio (Rogers and Yau, 2006). This inadvertently increases the radial vapour gradient between the particles' surfaces and the ambient vapour. As a result, the rate of diffusion of water vapour onto aerosol particles increases. As the saturation ratio nears

the saturation point, the condensational growth rate becomes steeper. This is the point on the curves where larger particles become unstable and grow exponentially as it is clearly marked by a distinct and wide separation between the first three and the rest of the aerosol size bins. Here the particles whose sizes are greater than the critical sizes of activation on the standard





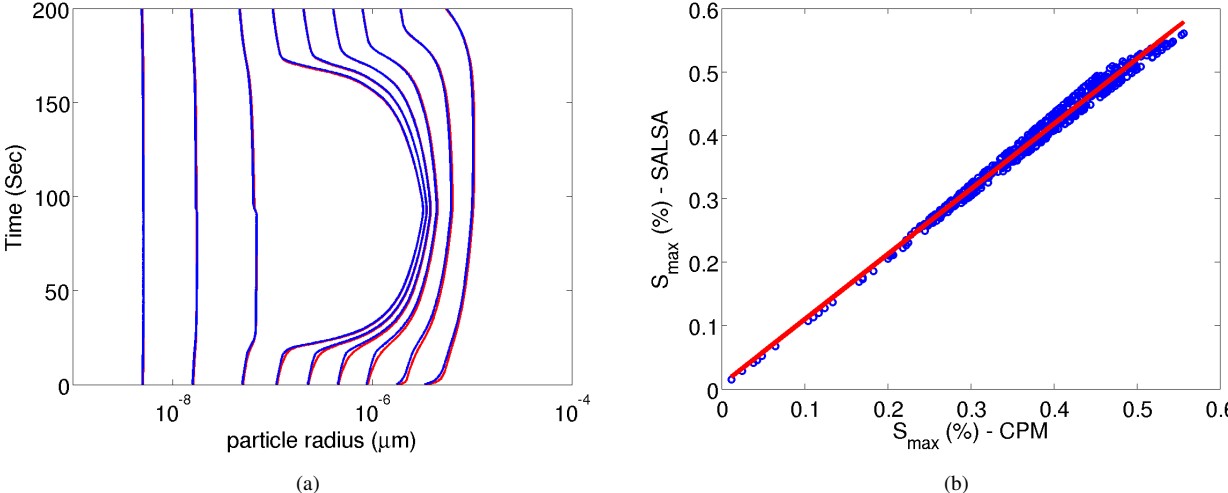

(a)
(b)

**Figure 1.** a) The ensemble mean of the evolution of the aerosol size distributions for both SALSA and the CPM from 1000 different model runs. The CPM is represented by red curves while SALSA is represented by blue curves. The ensembles were made by perturbing initial values of temperature and vertical velocity. b) The corresponding predicted maximum supersaturations from the same ensemble for cloud parcel model vs those from SALSA.

Kohler curve can be activated into cloud droplets (Kohler, 1936; Kokkola et al., 2003b; Petters and Kreidenweis, 2007) and they continue growing by condensation as long as the vapour gradient of the environment relative to the particle surface remains supersaturated. But it should be noted that the saturation ratio does not necessarily keep rising even if the parcel does, this is because any excess vapour is immediately dissipated by aerosol/cloud droplets thereby reducing the degree of supersaturation.

5    A distinct turning point is seen at around the 90 s timestep; this point marks the highest/coldest altitude reached by the air parcel after which the parcel starts descending and warming adiabatically and the droplets start evaporating. The parcel here follows the same temperature profile on its downward cycle and it is apparent that the curves closely show a symmetric shape, indicating the accuracy with which the models treat both condensation and evaporation processes. It can be seen that the evolution of the particle size distributions by SALSA is near perfect relative to those of the control model (the CPM) for

10   both condensation and evaporation; although in general, SALSA slightly under-predicts the growth especially of the larger size bins. Overall, we can conclude that the treatment of condensation and evaporation in SALSA is satisfactory. Even the predicted maximum supersaturations between the two models agreed with each other almost perfectly (Fig. 1b). Maximum supersaturation is an important parameter since it determines the population of interstitial aerosols that can be activated into cloud droplets in a given timestep.

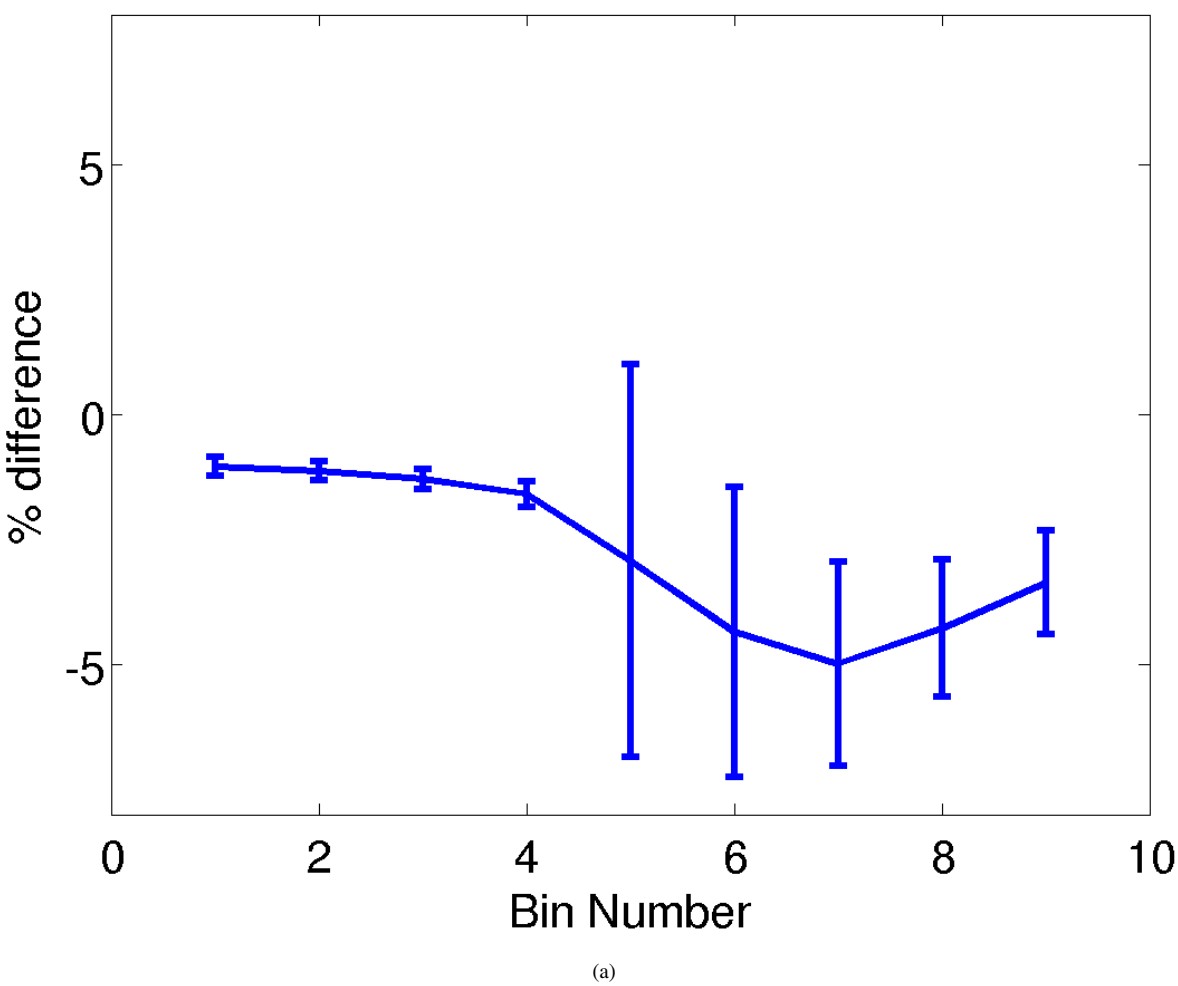

(a)

**Figure 2.** The relative % difference of the aerosol size distribution predicted by the CPM versus those predicted by SALSA derived from 100 different ensemble simulations. The ensembles were made by perturbing initial values of temperature, the relative humidity, the gas-phase concentrations of $HNO_3$ and $NH_3$ and their corresponding initial particle-phase concentrations.

### 3.2 Dissolution of $HNO_3$ and $NH_3$

The new dissolution routine that has been implemented in SALSA was also evaluated by comparing it with that of the CPM. The CPM uses the AIM model to treat the liquid phase thermodynamics hence for the evaluation of the APD in SALSA, we implemented the same thermodynamic model in order to reduce the degrees of freedom between the two models. The AIM

5   model is a mole-fraction-based thermodynamic model, which calculates the activity coefficients and the equilibrium partial





pressures for the $H_2O$-$HNO_3$-$NH_3$ system by using an iterative method for minimizing the Gibbs free energy (Clegg et al., 1998).

We again used the perturbed parameter ensemble approach as in the evaluation of the APC to create an ensemble of model outputs. We retained the same aerosol settings given in Table. 1; however, we slightly modified the initial chemical composition

of aerosol particles to include a fraction of $HNO_3$ and $NH_3$ at the beginning of the simulations. This modification was logical not only because most observed aerosol species are multicomponent in nature, but also because the gas-to-particle partitioning of a condensing compound depends on its particle-phase concentration. Here, we perturb the temperature, the relative humidity, the gas phase concentrations of $HNO_3$ and $NH_3$ and their corresponding initial particle-phase concentrations. Although the perturbations are random, there were upper and lower bounds for the perturbed variables. The temperature ranged between

263 and 272 K, RH ranged between 90 and 99 %, while the gas-phase concentrations of $HNO_3$ and $NH_3$ varied from 0.1 to 0.4 ppb and their corresponding particle-phase concentrations ranged between 25 and 60 % of the gas-phase concentrations. This was a closed system simulation in which there was no external replenishment of gas phase concentrations of both $HNO_3$ and $NH_3$. The model is run for 10000 s to allow the dissolution and the subsequent evaporation of $HNO_3$ from the particles to take place and to allow for equilibration of the particles. This model setup and experiment shall henceforth be referred to as

Ensemble B.

It should be noted that these dissolution simulations are performed isobarically and also, unlike condensation, which is more significant at higher relative humidities, dissolution of semi-volatile gases onto preexisting aerosols can occur at any relative humidity; although it is intensified at higher relative humidities and colder temperatures (Carslaw et al., 1994; Wang et al., 2016). It is also noteworthy that here we only tested the condensation/evaporation routine and the size classes do not match

exactly the ones used in SALSA by default. Instead they match those of the CPM.

Fig. 2 shows the relative % difference of the aerosol size distribution predicted by the CPM versus those predicted by SALSA. It is apparent that the two models compare with each other relatively well, especially in the lower tail end of the aerosol size distribution. The maximum % deviation between the two models is merely 5% and is seen in the mid-section of the aerosol size distribution. This satisfactory agreement in the treatment of dissolution is also noticeable in the particle-phase

concentrations of $HNO_3$ and $NH_3$ (figures not shown). The setup used in this test is relatively extreme since in the beginning, semivolatiles are introduced in the gas phase, which leads to quick uptake of these semivolatiles from the gas phase into the particle phase as was seen in another modelling study of Romakkaniemi et al. (2011), followed by slow equilibration towards equilibrium size distribution. In a 3-dimensional model setup, the changes are smoother and the expected performance would be even better.

### 3.3 The Effect of Dissolved Gases on Cloud Formation

The rationale behind implementing the dissolution of semi-volatile compounds in aerosol-cloud or climate models is that their presence in atmospheric aerosols potentially modifies the CCN activity of cloud nucleating aerosols (Kulmala et al., 1995; Kokkola et al., 2003b). In order to investigate this effect, we configured SALSA to nucleate cloud droplets and conducted two simulations with a slight modification to the setup described for the Ensemble A experiment. In this setup, only initial

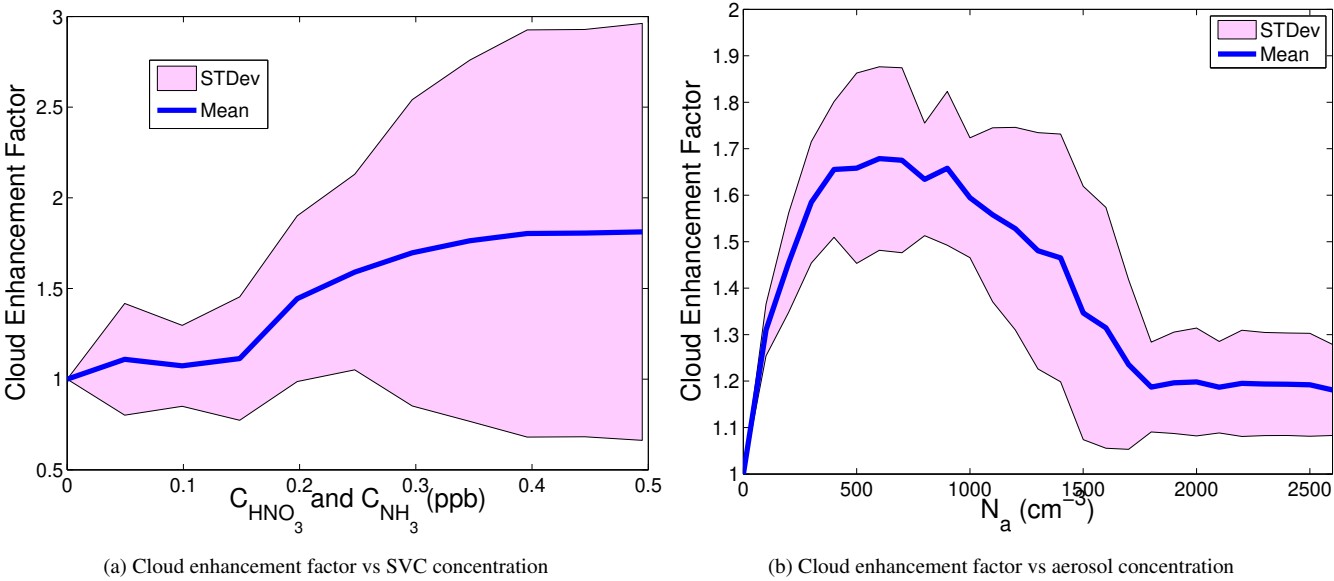

(a) Cloud enhancement factor vs SVC concentration        (b) Cloud enhancement factor vs aerosol concentration

**Figure 3.** (a) The dependence of the cloud enhancement factor on the change in SVC gas phase concentration for $170 \, \mathrm{cm}^{-3}$. (b) The dependence of the cloud enhancement factor on the change in background aerosol number concentration conducted for 0.5 ppb of SVCs. These ensembles were created using SALSA on a rising air parcel.

temperature, vertical velocities and initial particle-phase fractions of SVCs were randomly perturbed in order to create an ensemble of 100 simulations. In each ensemble member, the model was rerun 11 times with the only change in each rerun being the initial gas-phase concentrations of $HNO_3$ and $NH_3$. The initial gas-phase concentrations were increased stepwise from 0 ppb to 0.5 ppb, while the total number of cloud droplets formed during each rerun was recorded at the end of each

simulation. The aerosol concentration here was set at $170 \, \mathrm{cm}^{-3}$.

Consequently, an ensemble of cloud droplet enhancement factors, $CEF = \frac{N_{svc}}{N_0}$ was created, where, $N_0$ is the total number of cloud droplets formed when the gas-phase concentration of SVCs was 0 ppb (i.e the control simulation) and $N_{svc}$ was total number of cloud droplets formed in the presence of the increased gas-phase concentration of SVCs. Fig. 3a shows the response of the enhancement factors to increasing gas-phase concentrations of SVCs. A monotonic increase of $CEF$ with increasing

SVC concentrations is distinct. This implies that, under the tested atmospheric conditions, as the SVCs increase, more and more cloud droplets are nucleated. In a similar analysis by Kulmala et al. (1995), they found that for atmospheric aerosol concentrations of $110 \, \mathrm{cm}^{-3}$, the $CEF$ increased to 1.2 when $HNO_3$ concentrations were raised from 0.1 to 1 ppb and $CEF$ became 1.6 when $HNO_3$ was raised to 10 ppb. Their values of $CEF$ are slightly higher than the range of mean values shown in Fig. 3a.

It should however be noted that $CEF$ depends on other factors such as background aerosol concentration. In order to show this dependence, we repeated the above experiment, but instead of varying the initial gas-phase concentrations of SVCs, we kept SVC concentration constant at 0.5 ppb, while the number concentration of aerosol particles, $N_a$ was raised from 100

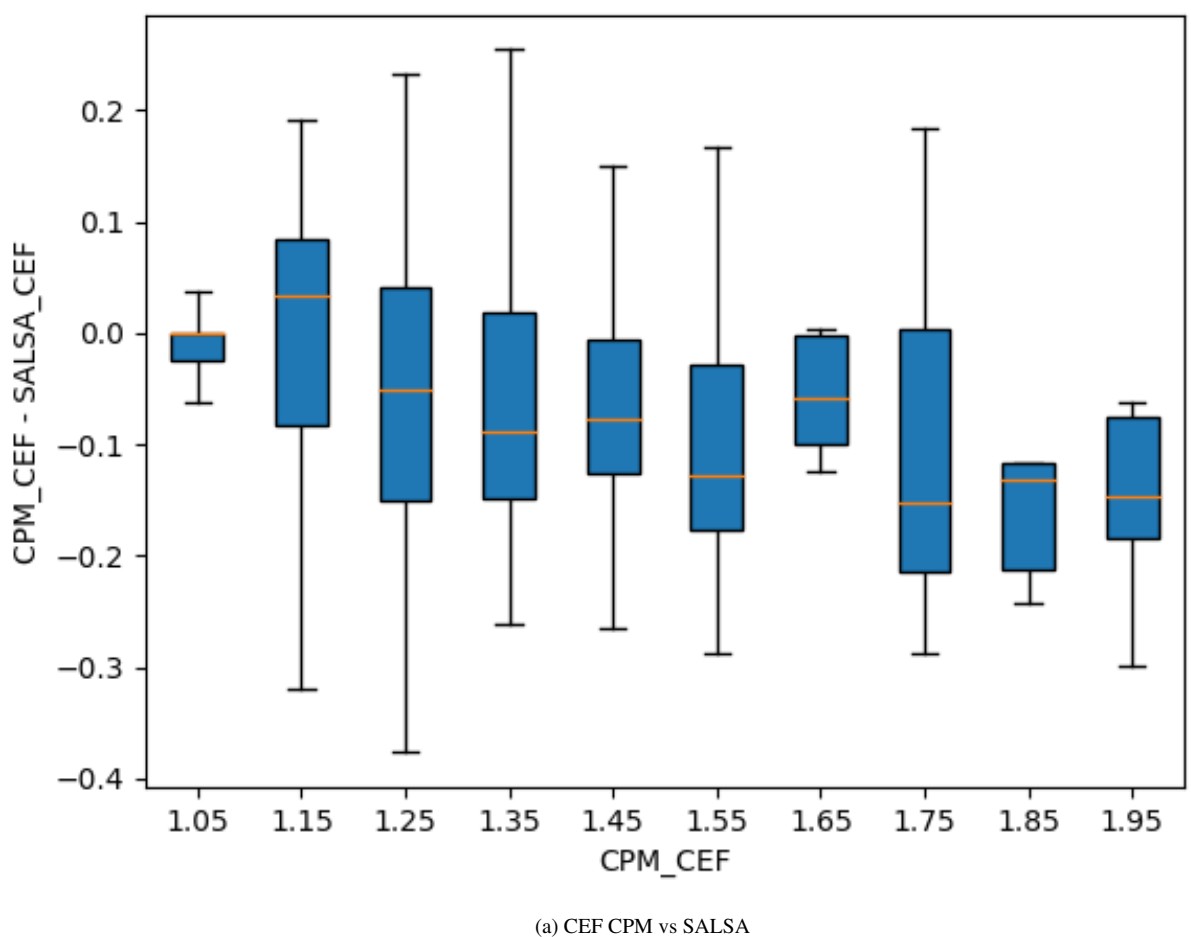

(a) CEF CPM vs SALSA

**Figure 4.** The cloud enhancement factor (CEF) for the CPM model plotted against the CEF for SALSA. The CPM model has 100 aerosol bins, while SALSA has 10 aerosol bins.

to 1500 $\mathrm{cm}^{-3}$. Fig. 3b shows that for a constant gas-phase concentration of SVC compounds, 0.5 ppb in this case, $CEF$ increases as $N_a$ is raised from 100 to 600 $\mathrm{cm}^{-3}$, which is where the maxima occurs. For aerosol concentrations above 600 $\mathrm{cm}^{-3}$, $CEF$ starts decreasing until it flattens out or saturates as more and more aerosols are added above 1800 $\mathrm{cm}^{-3}$.

In order to assess the performance of SALSA in accurately treating the dissolution process and translating that to the en-
5   hancement of the CCN activity of aerosols, we compared the corresponding cloud droplet enhancement factors (CEF) between SALSA and the CPM under the same atmospheric conditions. This was achieved by repeating the above experiment in which, initial temperature, vertical velocities, aerosol mean number diameter and modal number concentration were randomly perturbed in order to create an ensemble of 100 simulations and in each perturbation event, the SVCs were gradually increased





**Table 2.** The height dependent model initialization profiles. $z$ (**m**) is the altitude and $z_0 = 795$ **m** marks the lower level of the inversion layer.

| Thermodynamic quantity | $z < z_0$ | $z \geq z_0$ |
|---|---|---|
| Potential temperature, $\theta$ (K) | 283.3 | $295 - (z - z_0)^{1/3}$ |
| Mixing ratio $q_t$ (g kg$^{-1}$) | 9.45 | $3 - 5(1 - exp((z - z_0)/500))$ |
| u (m s$^{-1}$) | $3 + 4.3z/1000$ | $3 + 4.3z/1000$ |
| v (m s$^{-1}$) | $-9 + 5.6z/1000$ | $-9 + 5.6z/1000$ |

from 0.1 to 0.5 ppb in both models. However, in order to make the CPM more accurate, we increased the number of cloud bins from 10 to 100 bins, while the 10 size bins of the SALSA model were maintained. Fig. 4a is a box plot showing the absolute difference of the enhancement factors between the cloud parcel model and SALSA (CPM$_{CEF}$ against CPM$_{CEF}$ - SALSA$_{CEF}$), which shows a systematic overestimation of the CEF by the SALSA of at most -0.1 relative to CPM. This un-
derestimation is mainly attributed to the difference in the resolution of the size distribution of aerosols between the models. However, it has to be noted that even with 10 size bins covering the size range over three magnitudes, SALSA manages to reproduce the CPM results relatively well.

## 4   LES Simulations

To investigate, how HNO$_3$ and NH$_3$ affect the cloud properties in the full extent of a cloud when also other cloud processes
are active, we modelled cloud properties in a three dimensional LES simulation. In this section, we show the implementation of the new condensation-dissolution scheme in UCLALES-SALSA and analyze the behavior and effect of HNO$_3$ and NH$_3$ compounds on aerosols, clouds and precipitation properties. However, a more comprehensive analysis of the effects of these gases on the different microphysical processes and the effectiveness and pathways with which these gases are removed from the atmosphere in different meteorological and aerosol conditions shall be explored separately.
The treatment of semi-volatile gases in any LES or climate model is a huge improvement towards an accurate representation of aerosol-cloud processes; however, this implies additional computational burden and time on the host model. The dominant source of this extra computational expense emanates from the aerosol thermodynamic equilibrium module. When validating this model in 0-dimensional framework, we used the AIM thermodynamic module mainly because it contains the most comprehensive chemistry and thermodynamics than most of the other thermodynamic modules in use. From their comparative review,
Zhang et al. (2000) found AIM to be one of the most accurate thermodynamic models. However, it is computationally very expensive owing to the iterative approach used in its calculations, especially for low relative humidities and hence rendering it unfavourable for large-scale or operational applications. Therefore, for the LES simulations we switched onto the ISORROPIA module (Nenes et al., 1998; Fountoukis and Nenes, 2007), which has a comparable accuracy relative to the AIM module.
One additional change was made to SALSA when ISORROPIA was coupled to it. When using ISORROPIA, we noted
that it over-predicts the gas-phase equilibrium mole concentrations of nitric acid, $C_{HNO_3}$ at very high relative humidities (i.e





**Table 3.** The distribution parameters of the bimodal log-normal distribution assumed in UCLALES-SALSA for the DYCOMS-II simulations. The parameters were taken from Asmi et al. (2011) for the for the Hyytiälä (SMR) measurement station, which is a forestry field station located 200 km north of Helsinki, Finland.

| Parameter | $1^{st}$ mode | $2^{nd}$ mode |
|---|---|---|
| Number concentration, $\mathrm{cm}^{-3}$ | 1008 | 144 |
| Geometric mean, μm | 0.057 | 0.201 |
| Standard deviation ($\sigma$) | 1.96 | 1.36 |

when $RH > 99.5\%$). The implication of this over-prediction was that the partitioning of $HNO_3$ into the particle-phase was then suppressed at these humidities. This is because ISORROPIA is originally not designed to operate in such high relative humidity conditions (Nenes et al., 1998). Therefore, in order to circumvent this discrepancy, we modified the thermodynamic model so that it predicts the saturation vapor concentrations of $C_{HNO_3}$ and $NH_3$ directly from the Henry's law where relative

humidity exceeded %. This was a physically rational modification because at such high RH, the droplets are highly dilute and condensing compounds can be assumed to follow near ideal behaviour.

### 4.1 Case Description and Model Configuration

The case simulated here is the DYCOMS-II, flight RF02 case of Stevens et al. (2003), which was conducted off the west coast of Califonia in July 2001. This is the same case against which the original version of the UCLALES-SALSA model was robustly

validated at its inception (Tonttila et al., 2017). The only relevant major microphysical update that has been implemented since then is in the condensation and dissolution routine whose validation has been presented in the previous sections. Consequently, there is no detailed discussion related to its validation and performance on the simulated case.

Briefly, the simulations in UCLALES-SALSA are initialised using vertical profiles of potential temperature, $\theta$, total mixing ratio of water vapour, $q_t$, and the $u$ and the $v$ components of horizontal wind and for this case, they are described by height

dependent mathematical expressions given in Table 2. The simulation domain was $5 \times 5$ km in total with the horizontal resolution set at 50 m for both the x and y directions, while the vertical resolution was 20 m with the domain top set at 1.6 km. A 200 m thick, rigid damping layer is placed between 1.4 and 1.6 km levels in order to prevent gravity waves from reflecting and amplifying into the simulation domain. The integration time step in the model is dynamical, (i.e. not fixed) during the simulations; however, the maximum model time step allowed in these simulations was 1 s. The aerosol properties applied for the

control simulations are given in Table 3. Although the meteorological conditions are for the marine stratocumulus, we employ continental aerosol as it produces more realistic background to demonstrate the potential effect of inorganic semivolatiles on cloud microphysics. The simulation duration was 12 hours and the output was written out every 60 s for analysis. A model spin-up time of 1 hour was allowed for the turbulence development during which rain formation and coagulation processes were switched-off, and the second hour is allowed for initialization of drizzle. The results are thus presented after two hours

into the simulation.





## 4.2 Experiment Set-up

Two three-dimensional simulations were performed to examine the interaction of semi-volatile compounds with aerosols, clouds and precipitation. The control simulation (CTRL) was run without $HNO_3$ and $NH_3$ in the gas-phase, although a fraction of $NH_3$ was available in the particle-phase in order to neutralise the sulphate (i.e. 20 % of the volume of the initial particle

phase was $NH_3$, while 80 % was sulphate). The second simulation was a closed system simulation (in terms of gas-phase concentrations of semi-volatile compounds) in which 1 ppb of $HNO_3$ and 1 ppb of $NH_3$ were introduced in the gas-phase only at the beginning of the simulation. These are the typical concentrations for an urban scenario. This simulation shall be referred to as the SV simulation. Other thermodynamic and meteorological conditions and model configurations remained identical between the two simulations.

## 4.3   Results and Discussion

Fig. 5 presents time-height plots showing the time evolution of the cloud droplet number concentrations (CDNC), the mean droplet sizes and the rain rate for the control simulation in the left column and for the SV simulation in the middle panel. There is a notable increase in CDNC, which happens because the dissolved gases increase the sizes especially those of interstitial aerosols, which then reduces their corresponding critical saturation ratios (Kokkola et al., 2003b), and hence increasing the

number of aerosols that can be activated into cloud droplets. There are currently no coincident measurements available against which we could compare this fractional increase in droplet concentration due to uptake of semivolatiles. However, our fractional increase of about a factor of about 20% for 1 ppb of $HNO_3$ and $NH_3$ is within the same order of magnitude with those found by Kulmala et al. (1995) who predicted a factor of 1.2 increase for a 1 ppb gas-phase $HNO_3$ concentration, although their background aerosol concentration was relatively cleaner. Of course, other conditions upon which the experiments were

conducted are different but the results compare reasonably well.

There are several bearings of this increase in cloud droplet numbers that may affect the radiative, micro-physical and macro-physical properties of clouds. But ordinarily, the immediate effect is that the droplet sizes are decreased, owing to increased competition for available water vapor by extra cloud droplets and decreased supersaturation in cloud updrafts. This can be also seen in the results. However, for certain size ranges, condensation of SVCs may also increase the size of some droplets

as it increases the amount of solute, and this could be seen as a change in relative dispersion of cloud droplet size distribution (Xue et al., 2014). In our simulation the inherent consequence of a reduction in cloud droplet sizes, is the reduction in the ability of that cloud to produce precipitation. This is because the dominant mechanism through which precipitation is produced from the the clouds is collision and coalescence, which is heavily dependent on the size of colliding cloud droplets. Within UCLALES-SALSA this process is explicitly represented. Although the overall precipitation in the simulated case is

low, this clearly shows a distinct reduction in precipitation in the presence of SV compounds, which is consistent with what is expected. As the amount of formed precipitation is fairly low, the dynamical feedbacks due to changes in liquid water content and precipitation evaporation are too small to be analyzed.







(a)

**Figure 5.** The results from the simulations for the domain average cloud droplet number concentration, the cloud droplet mean size, and the rain rate as a function if time. On the left without semivolatile gases, in the middle with 1 ppb of $HNO_3$ and $NH_3$, and on the right ratio of presented properties

Fig. 6 shows the domain averaged In order to understand the physical transformations of $HNO_3$ and $NH_3$ across the different physical states and hydrometeor species, we conducted a mass budget analysis of these species. distribution of semivolatile gases between different phases. It is obvious that the dissolution of nitric acid is strongly depended on relative humidity as opposed to ammonia, which immediately dissolves to neutralize sulphate aerosols in a manner seemingly independent of

5   ordinary ambient conditions and does not easily evaporate once it dissolves into the particle-phase. It is apparent that much of the nitric acid stays in the gas phase especially under sub-saturated conditions. Within saturated or at very high RH regions (e.g. the cloud deck) however; almost all the gas-phase $HNO_3$ is dissolved into aerosol particles. Much of this dissolved $HNO_3$ is





(a)

**Figure 6.** The time evolution of domain averages of (a) Gas-phase concentrations and (b) particle-phase concentrations in aerosols (c) droplet concentrations and (d) drizzle for $HNO_3$ on the left and $NH_3$ ammonia on the right in the SV-1ppb simulation. The color scale is limited in the lower end to make differences visible. In the drizzle contains small amounts of nitrate whenever drizzle is available.

immediately transferred into cloud droplets when the hosting aerosol particles have been activated into cloud droplets. It therefore appears in this case that only trace amounts of nitrate would remain in the aerosol-phase relative to the cloud phases. A relatively small fraction of SVC would be found in the drizzle-phase mainly because precipitation water content is always much lower compared to cloud water content in the simulated cloud. However, as precipitation falls out of the cloud, much of
5   this nitrate is likely to evaporate back into the gas phase when droplets evaporate.



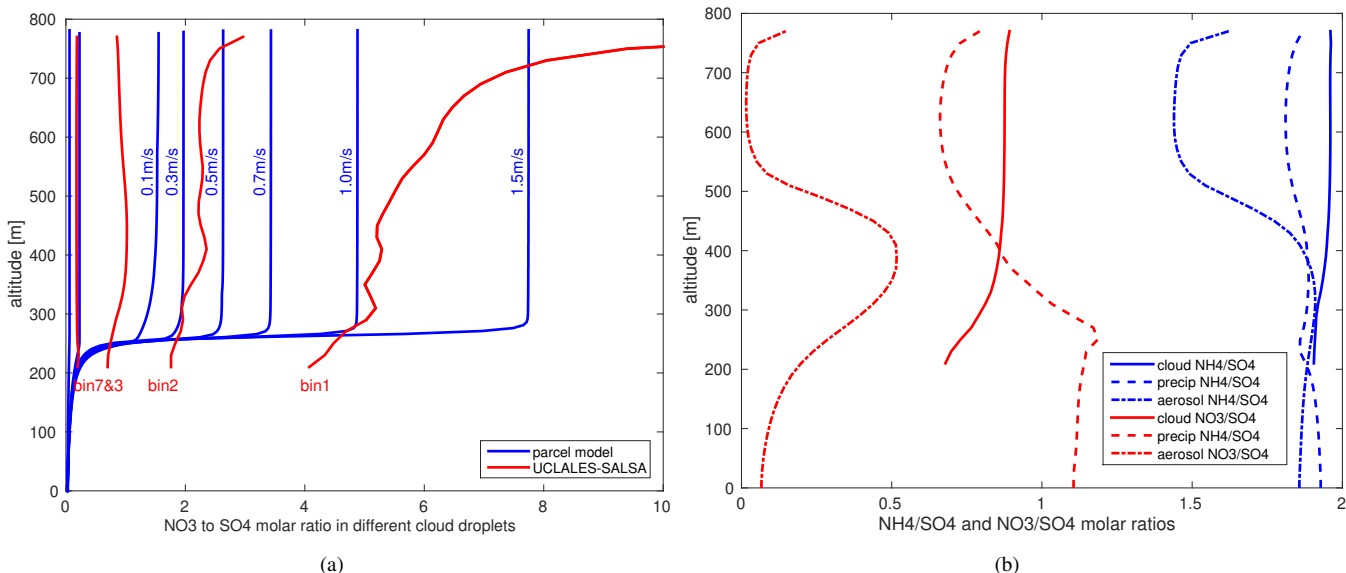

(a)                                                (b)

**Figure 7.** Relative partitioning of the semivolatiles in different hydrometeor species. a) Ratios of $NO_3^-$ to $SO_4^{2-}$ in different sized droplets for UCLALES-SALSA (red) compared to the ratios simulated with the cloud parcel model (blue) using different updraft velocities. The bins labelled in red correspond to the cloud bins in UCLALES-SALSA. b) The composition of different hydrometeor species (aerosol, cloud and drizzling droplet).

As with nitrate, the dissolved $NH_3$ will be transferred into cloud droplets and subsequently onto precipitation droplets if grown big enough. However, the fraction of ammonia that progresses into precipitation droplets will likely not evaporate when precipitation evaporates, but stays in the cloud processed aerosol to neutralize sulphate.

In order to take a closer look on the partitioning of different chemical compounds, in Fig. 7 a) we are presenting the ratio of
$NO_3^-$ to $SO_4^{2-}$ in different sized droplets during the last simulated hour with UCLALES-SALSA, and compare that to the ratio simulated with the cloud parcel model using different updraft velocities. From the figure, we can see that UCLALES-SALSA is capable of simulating the nitrate enrichment in the smallest activated droplet sizes, which is seen in cloud parcel simulations (Kokkola et al., 2003a). With the used input gas phase composition and aerosol size distribution, an updraft velocity of $0.8\,\mathrm{m/s}$ or higher is needed for the activation of largest particles in SALSA bin number 4 (cloud bin 1), corresponding closely on the
average cloud bin 1 composition at the cloud base. With increasing updraft velocity of a cloud parcel, it is more and more likely that the parcel penetrates higher in the cloud, increasing the average $NO_3^-/SO_4^{2-}$ ratio as a function of height in UCLALES-SALSA. Very close to the cloud top even stronger increase in $NO_3^-$ fraction can be observed. In the simulation setup, a constant gas phase concentration profile was used as an input, which can be seen as an entrainment of semivolatile gases from the cloud top. These gases are quickly uptaken by the smallest cloud droplets. Fastest entrainment is connected to strongest updraft
velocities of air parcels, highlighting the effect of nitrate enrichment at the cloud top (Fig 7a). However, such input profile for gas phase species might be unrealistic and thus the effect is not expected to be observed.





Because the different sized droplets have different composition, it can be expected that there is some difference also between the composition of aerosol, cloud and drizzling droplets. This is analyzed further in Figure 7 b). Again, UCLALES-SALSA is reproducing what can be expected of the composition of different hydrometeor classes. Below cloud, the fraction of nitrate remains fairly low until cloud formation after which nitrate condenses efficiently in cloud and precipitation droplets. Within the

cloud layer, the fraction of nitrate is clearly higher in cloud and precipitation droplets. In the aerosol particles, higher fractions of nitrate are seen at the cloud base altitude which varies between approx 200-500 m. Interestingly, we can also observe that interstitial aerosol is more acidic than aerosol below the cloud. This is caused by nitric acid uptake on cloud droplets, followed by ammonia neutralization through gas-phase transition from aerosol to cloud droplets similar to what was also observed in field measurements (Hao et al., 2013). In UCLALES-SALSA, drizzle is formed through collision between cloud droplets, and

thus the larger droplets are more likely to form drizzle. Thus within the cloud layer, the formed drizzle contains less nitrate and ammonia than the average cloud droplet. After drizzling below the cloud base, droplets will uptake nitric acid from the gas phase, and thus increasing the $NO_3^-/SO_4^{2-}$ ratio. In the drizzle reaching the ground $NO_3^-/SO_4^{2-}$ ratio is actually larger than in the cloud droplets which means that wet removal is relatively more efficient for nitrate than for sulfate. For ammonia this effect of uptake of gas phase ammonia to precipitating droplets is weaker at least in this simulation setup.

**5   Conclusions**

The co-condensation of water vapour and inorganic semi-volatile gases was implemented in our state-of-the-art LES model - UCLALES-SALSA, and we investigated how the gases are partitioned between gaseous and particle phases. The condensation of water vapour onto pre-existing aerosol particles and hydrometeors was treated using the analytical predictor of condensation scheme, while the deposition of SVCs was treated using the analytical predictor of dissolution. Both of these techniques

are semi-implicit techniques of Jacobson (2005) used to solve the growth equations. It was noted that under sub-saturated conditions, the APC scheme was unstable in treating the condensation of water vapour - it produced spurious oscillations. As a result, the APD was used instead and it evened out these oscillations.

The validation of the new deposition and condensation scheme was performed based on a more detailed cloud parcel model (CPM) due to the absence of measurements dedicated for these processes. The comparison between the CPM and SALSA

showed a near perfect agreement between the two models for both condensation/evaporation and dissolution. On the average, SALSA exhibited a slight underestimation of dissolved compounds, which could be attributed to the coarser time-steps and a more simplified method for solving condensation equations used in SALSA than in the CPM. These validation experiments were conducted on an air-parcel that was raised from 1000 hPa to above the condensation level and descended back to the original altitude in order to ensure that both condensation/dissolution and evaporation of water vapour and SVCs are tested

under different humidity conditions.

The effect of SVCs on the CCN activity of aerosols was explored using SALSA in a box model approach in which an air parcel comprising of about 1000 cm$^{-3}$ of aerosols was raised to altitudes above the condensation level. It is apparent that as the SVCs gas-phase concentration was increased from 0 to 0.5 ppb, the CCN activity as indicated by the mean cloud enhancement





factor (CEF) rose gradually to as much as $1.2 \pm 0.1$. In addition, it was also shown that the CCN activity for a given SVC gas-phase concentration is also dependent on the number concentration of background aerosols. For instance, for a constant concentration of 0.5 ppb of SVCs, the cloud enhancement factor increased with increasing aerosol concentration until a peak is reached, after which it starts decreasing with increasing particle concentration before it saturated as the number of background

aerosols was raised from 100 to 1400 $cm^{-3}$ - the optimum background aerosol concentration with the highest CEF was about 600 $cm^{-3}$ and it saturated higher concentrations above 2000 $cm^{-3}$. This indicates that the the influence of SVCs on the CCN activity is controlled by several atmospheric factors, which may also include other ambient meteorological conditions such as temperature and RH.

The typical concentrations of $HNO_3$ and $NH_3$ in the atmosphere are about 0.1 - 1 ppb, but these can be up to 50 ppb in

highly polluted environments. We tested the effect of these SVCs in both zero- and three-dimensional simulations using an LES and 1 ppb concentration of $HNO_3$ and $NH_3$ in the gas phase at the beginning of the simulation without replenishing them during the course of the simulation. The effect of SVC concentrations on cloud formation and properties was found to be similar to that caused by the Twomey effect, in the sense that a significant increase in droplet number concentration was predicted in the presence of SVCs relative to the control simulation. The overall amount of precipitation was small in

the simulated stratocumulus case, hence the increase in droplet count led to a smaller mean cloud droplet sizes and reduced drizzle. In a population of cloud droplets with a wide spectrum of size distribution, the smaller droplets contain relatively higher amounts of nitrate than the larger ones, and hence, as the drizzle is mainly formed through large droplets, the scavenging of ammonium nitrate in clouds is weaker than would be estimated because smaller droplets are poorly collected poorly eliminated during in-cloud coagulation and collection processes. The model was also able to simulate the relatively more acidic interstitial

particles than cloud droplets. However, below the cloud, condensation of gases on drizzling droplets quickly increases their overall wet scavenging efficiency compared to sulphate.

This model framework provides a platform on investigating the effect of SVCs on cloud formation and further on, their effect on aerosol forcing. It can also be used to parameterize the effect of SVCs on cloud formation for use in large scale aerosol models, such as chemical transport models and climate models.

*Code availability.* The model source code and input files needed to reproduce the simulations presented in this paper can be downloaded from Github at https://github.com/UCLALES-SALSA/UCLALES-SALSA_v1.1 with a DOI: https://zenodo.org/badge/doi/10.5281/zenodo.3982709.svg.

*Author contributions.* The first author conducted the model development and validation particularly the condensation and dissolution routines together with the experiments and analysis of the results. Juha Tonttila and Tomi Raatikainen participated in model development mainly

of the other elements of UCLALES-SALSA and interpretation of the results, while Sami Romakkaniemi and Harri Kokkola directed and guided the research work. All authors also actively participated in the writing of the manuscript.





*Acknowledgements.* This work is supported by the Academy of Finland (project nos 283031 and 322532 and the Centre of Excellence in Atmospheric Science, no. 272041), the European Research Council (Consolidator Grant 646857), Horizon 2020 Research and Innovation Programme under Grant Agreement 821205 (FORCeS), and Tiina and Antti Herlin foundation (grant number 20200025).



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
