# Peer review of "Implementing Gas-to-Particle Partitioning of Semi-Volatile Inorganic Compounds in UCLALES-SALSA"

_Atmospheric Chemistry and Physics, 2020_

## Referee Comment (RC1) · Anonymous Referee #1 · 22 Nov 2020

General Comments

This paper presents an implementation of the process of dissolution of semi-volatile inorganic gases (HNO3 and NH3) into aerosol/cloud/precipitation within their sectional microphysics model coupled with a large-eddy simulator. Subsequently, investigated are the impacts on aerosol size distribution, cloud droplet numbers and size, and rain rate.

The context of this paper is topical, especially due to anticipated increase in ammonia and nitric acid, especially over polluted regions, and furthermore, the consequent impacts on new particle formation. However in its present form it fails to contribute

substantially to our scientific knowledge (Kulmala et al., 1993; Laaksonen et al., 1998, Kokkola et al., 2003, Romakkaniemi et al., 2005 – are important examples in literature by some of the present co-authors themselves, among various other papers), by being simply an implementation of Jacobson, 1997 into Tonttila et al., 2017. Another issue with this paper, as the title itself portends, is that it reads more like a technical report than a scientific paper. It may be that it was initially drafted towards publication in GMD and thus model (incremental) development and validation are provided more focus than unraveling substantial scientific points. The authors themselves note this in P14L12: "However, a more comprehensive analysis of the effects of these gases on the different microphysical processes and the effectiveness and pathways with which these gases are removed from the atmosphere in different meteorological and aerosol conditions shall be explored separately." Furthermore, the draft appears rushed, lacking goal-oriented structuring, with incomplete sentences and errors (detailed in following sections). In addition to the above, the consideration of co-condensation of semi-volatile inorganic gases and associated water uptake is not even a novel proposition. While the authors purport in P03L05 that they "assess for the first time, the efficiency with which these semi-volatile gases are sequestered from the atmosphere", the bulk of this paper is focused elsewhere, and it is not the "first time" (a recent example is Luo et al., 2019 & 2020). For these weaknesses and considering the scope of ACP, which carries high-quality studies that further our understanding of atmospheric chemical and physical processes, my recommendation is to reject the present paper for publication.

Specific Comments

Section 2.3.1 and 2.3.2 can go into an appendix as they are not original developments (Jacobson, 1997, 2002, 2005).

Section 3.1: Kokkola et al., 2018 have already developed confidence in the SALSA condensation routine. What is new here?

P11L33: "we configured SALSA to nucleate cloud droplets"; isn't this the parameterizations of Abdul-Razzak and Ghan in SALSA?

Definition of CEF: What are the assumptions and resultant errors associated with using a simple ratio of CDNsvc:CDN0 as a quantitative estimator of the effect on cloud formation? Or in other words, in Fig. 3(a), CEF = 1 falls within $\mu \pm \sigma$ of CEF w.r.t. [SVC]; is there then a (demonstrated) dependence?

Section 3.3: The discussion here is merely descriptive and does not provide insight.

Section 4.2: Why, if these (P16L06: "1 ppb of HNO3 and 1 ppb of NH3") values of SVCs were relevant, were they not in the range of evaluation of SALSA-standalone?

P16L12: "There is a notable increase in CDNC, which happens because the dissolved gases increase the sizes especially those of interstitial aerosols, which then reduces their corresponding critical saturation ratios (Kokkola et al., 2003b), and hence increasing the number of aerosols that can be activated into cloud droplets." What and how notable is the increase in CDNC? Where and by how much is the increase in aerosol sizes? Kokkola et al., 2003b is not needed here. What is the ostensible increase in CCN? What the reader would be interested in is a discussion supported by quantitative/statistical values.

P16L19: "Of course, other conditions upon which the experiments were conducted are different but the results compare reasonably well." Yet again is a statement that appears ambiguous due to the absence of details/numbers.

A general issue throughout the paper is that the write-up associated with the figures are descriptive in nature. While this is alright to guide the reader through the figures, the absence of a quantitative discussion (things that answer questions such as "How much more/less?", "How significant?", "Why/How much contribution of purported cause-effect?") makes the discussion appear speculative. Furthermore, a detailed discussion supported by statistics may uncover insights/interesting scientific points that are not currently demonstrated.

Technical Corrections

Table 1: Please recheck the values of the mean and standard deviation of particle diameters. There seems to be a copy-paste error. Additionally, standard deviation has the same units of measurement as the variable.

Figure 1: (a) x-axis: Confirm if radius or diameter. Also, the units reported seem wrong.

Figure 4: There is a panel label for the figure which is unnecessary and presently mislabeled.

Figure 5: It is unclear how the ratios (right panels) can go below 0. Also, the units reported for precipitation rate is W/m2; is this the cooling due to precipitation or change in clear-sky radiative cooling or just an error and should be kg/s/mˆ2?

Figure 7: Please recheck the legend of panel (b).

P01L15-24: References can be cleaned up: either they may be omitted if the above is cited, since all the information has moved from literature to textbooks, or the more pertinent and initial studies should be cited.

P01L17: IPCC AR5, 2013 would be a better "scientific" reference than Pörtner et al., 2019, which, although more recent, is a summary for policymakers.

P02L21: Please add a line about changing atmospheric ammonia for completeness.

P03L13: Period after "Section" to be removed. Also "the last section of the paper" to be replaced with "Section 5".

P14L09: Please remove the comma.

P17L01-02: "Fig. 6 shows the domain averaged" is out of place.

P17L03: "depended" -> "dependent"

P17L03: "It is obvious that. . ." It may not be obvious to the reader. Please remove these words and then discuss why is it so.

P17L05: "It is apparent that..." It may not be apparent to the reader. Please remove these words and then discuss why is this so.

References

IPCC AR5: Climate Change 2013: The Physical Science Basis. Contribution of Working Group I to the Fifth Assessment Report of the Intergovernmental Panel on Climate Change, Cambridge University Press, Cambridge, United Kingdom and New York, NY, USA, 2013.

Jacobson, M. Z. (1997). Numerical techniques to solve condensational and dissolutional growth equations when growth is coupled to reversible reactions. Aerosol Science and Technology, 27(4), 491-498.

Kokkola, H., Romakkaniemi, S., and Laaksonen, A.: Köhler theory for a polydisperse droplet population in the presence of a soluble trace gas, and an application to stratospheric STS droplet growth, Atmos. Chem. Phys., 3, 2139–2146, doi:10.5194/acp-3-2139-2003, 2003.

Kokkola, H., Kühn, T., Laakso, A., Bergman, T., Lehtinen, K. E. J., Mielonen, T., Arola, A., Stadtler, S., Korhonen, H., Ferrachat, S., Lohmann, U., Neubauer, D., Tegen, I., Siegenthaler-Le Drian, C., Schultz, M. G., Bey, I., Stier, P., Daskalakis, N., Heald, C. L., and Romakkaniemi, S.: SALSA2.0: The sectional aerosol module of the aerosol–chemistry–climate model ECHAM6.3.0-HAM2.3-MOZ1.0, Geosci. Model Dev., 11, 3833–3863, https://doi.org/10.5194/gmd-11-3833-2018, 2018.

Kulmala, M., Laaksonen, A., Korhonen, P., Vesala, T., Ahonen, T., and Barrett, J. C.: The effect of atmospheric nitric acid vapor on cloud condensation nucleus activation, J. Geophys. Res., 98, 22949–22958, 1993.

Laaksonen, A., Korhonen, P., Kulmala, M. and Charlson, R. J.: Modification of the Köhler equation to include soluble trace gases and slightly soluble substances, J. Atmos. Sci., 55, 853– 862, 1998.

[Figure]

Luo, G., Yu, F., and Moch, J. M.: Further improvement of wet process treatments in GEOS-Chem v12.6.0: impact on global distributions of aerosols and aerosol precursors, Geosci. Model Dev., 13, 2879–2903, https://doi.org/10.5194/gmd-13-2879-2020, 2020.

Luo, G., Yu, F., and Schwab, J.: Revised treatment of wet scavenging processes dramatically improves GEOS-Chem 12.0.0 simulations of surface nitric acid, nitrate, and ammonium over the United States, Geosci. Model Dev., 12, 3439–3447, https://doi.org/10.5194/gmd-12-3439-2019, 2019.

Romakkaniemi, S., Kokkola, H., and Laaksonen, A.: Soluble trace gas effect on cloud condensation nuclei activation: Influence of initial equilibration on cloud model results, J. Geophys. Res., 110, D15202, doi:10.1029/2004JD005364, 2005.

Tonttila, J., Maalick, Z., Raatikainen, T., Kokkola, H., Kühn, T., and Romakkaniemi, S.: UCLALES–SALSA v1.0: a large-eddy model with interactive sectional microphysics for aerosol, clouds and precipitation, Geosci. Model Dev., 10, 169–188, https://doi.org/10.5194/gmd-10-169-2017, 2017.
* * *

---

## Referee Comment (RC2) · Anonymous Referee #2 · 2 Dec 2020

The authors present a study in which they have developed a large eddy simulator to include the condensation of inorganic semi-volatile compounds (SVCs) condensing on a particle population to assess the impact it has on cloud droplet formation and subsequently cloud microphysics. Owing to the current lack of an appropriate data set, the LES predictions are evaluated (favorably) against those acquired from cloud parcel model simulations which allow for a higher particle size resolution.

The authors find the expected result that cloud droplet nucleating potential is increased by condensation of these SVCs due to increased particle size/soluble mass, and consequently a cloud droplet spectrum with reduced mean size and therefore precipita-

tion. In this regard the paper does not present a serious lurch forward in understanding these processes/mechanisms. It would have been nice to see with the inclusion of condensing semi-volatile organic compounds in a VBS framework simultaneously, so as to ascertain whether or not there would be any interesting interaction effects, though most likely just a compounded effect on CCN activity.

At present I do not see that the paper is suitable for publication in ACP as it does not contain any new scientific insight. That said, it is nice to see these processes included in an LES and a more substantial study would certainly benefit from the increased process resolution.

Comments: Figure 2: Would be useful to clarify in the caption/y-axis that this is the change in bin radius (as opposed to number concentration) Figure 5: mean droplet sizes less than 0? Figure 7a: Could clarify in the caption that the parcel model curve correspond to "total" molar ratio Figure 3: Would be interesting to see the total variability, i.e. min/max, as well. Also, might be worth noting in the text that the CEF is negative in some cases and elaborate why this might be.

---

## Author Comment (AC1) · 21 Dec 2020

We thank both reviewers for the thorough reviews and comments on the manuscript. It is clear that we have not clearly written out the novelty of this study and this is why the reviewers are suggesting the paper not to be published. However, as detailed below, there are several aspects in this study that go well beyond what has previously been done in research related to the aerosol-cloud interactions of semivolatile compounds, especially ammonium nitrate. We are certain that elaborating the novelty in our approach and extending the analysis on the process level will improve the manuscript and make it interesting for the scientific community working on aerosol-cloud interactions.

**1 Referee 1**

**1.1 General Comments**

– This paper presents an implementation of the process of dissolution of semi-volatile inorganic gases ($HNO_3$ and $NH_3$) into aerosol/cloud/precipitation within their sectional microphysics model coupled with a large-eddy simulator. Subsequently, investigated are the impacts on aerosol size distribution, cloud droplet numbers and size, and rain rate. The context of this paper is topical, especially due to anticipated increase in ammonia and nitric acid, especially over polluted regions, and furthermore, the consequent impacts on new particle formation. However in its present form it fails to contribute substantially to our scientific knowledge (Kulmala et al., 1993; Laaksonen et al., 1998, Kokkola et al., 2003, Romakkaniemi et al., 2005 – are important examples in literature by some of the present co-authors themselves, among various other papers), by being simply an implementation of Jacobson, 1997 into Tonttila et al., 2017. Another issue with this paper, as the title itself portends, is that it reads more like a technical report than a scientific paper. It may be that it was initially drafted towards publication in GMD and thus model (incremental) development and validation are provided more focus than unraveling substantial scientific points. The authors themselves note this in P14L12: "However, a more comprehensive analysis of the effects of these gases on the different microphysical processes and the effectiveness and pathways with which these gases are removed from the atmosphere in different meteorological and aerosol conditions shall be explored separately." Furthermore, the draft appears rushed, lacking goal-oriented structuring, with incomplete sentences and errors (detailed in following sections). In addition to the above, the consideration of co-condensation of semi-volatile inorganic gases and associated water uptake is not even a novel proposition. While the authors purport in P03L05 that they "assess for the first time, the efficiency with which these semi-volatile gases are sequestered from the atmosphere", the bulk of this paper is focused elsewhere, and it is not the "first time" (a recent example is Luo et al., 2019 and 2020). For these weaknesses and considering the scope of ACP, which carries high-quality studies that further our understanding of atmospheric chemical and physical processes, my recommendation is to reject the present paper for publication.

We are grateful to the referee for the insightful comments, which we agree on and feel that they are to a large extent fair. However, we feel that their final recommendation to reject the paper is a bit too harsh given or based on their comments about the work and the manuscript. Kindly find below our arguments on why we think the recommendation is too harsh.

The addition of Jacobson et al. (1996) scheme into Tonttila et al. (2017) is not the main thrust of this work, although it is an integral component. Typically to solve the size dependent condensation of semivolatile acids and bases on different sized particles, an ordinary differential equation solver employing specific methods, is needed. Such solvers are efficient in a box model framework as the solution is highly accurate. However, for a set of stiff differential equations such solvers are computationally too expensive to be used in 3D applications. This is the main reason why atmospheric models typically employ either assumption on aerosol bulk composition or equilibrium between aerosol and gas phases. To the best of our knowledge of the literature, no other currently available scheme provides a size-resolved dynamic treatment of the partitioning of ammonia and nitrate in aerosol-cloud schemes that is computationally feasible in fully-fledged 3-dimensional modelling frameworks such as UCLALES-SALSA, which we achieved by a novel solution to semi-explicit integration of the dissolution equations. The employment of Jacobson scheme is not as straightforward as may be assumed, and we acknowledge that we failed in the motivation for representing all the details of how the new model agrees with a more accurate solution. Out-of-box employment of the APC and the APD equations is not working even in the case of water condensation for the formation/evaporation cycle of cloud droplets, and even less when condensation of all different components is solved simultaneously. The problem is that using APC and APD as they are, even with 1 second time step there will be oscillatory behaviour. On one time step, all nitrate will condense in the particle phase and on the next it will evaporate. This behaviour is amplified by simultaneous oscillation of water. Here we circumvented this problem by applying APD also for water. APD (an analytical predictor of dissolution) is meant to be used for dissolving species but here we apply the equations for water by obtaining the "effective Henry's law constant" (see $H'$ in Equation (5) in the manuscript) from the aerosol thermodynamics model. This removes the oscillatory behaviour in relative humidities $< 100\%$.

The recent references that the referee cited are quite important but they were unfortunately missed in the current pre-print; however, their treatments differ fundamentally with ours in detail and depth. The treatments of Luo et al. (2019, 2020) used to a larger extent empirically-based parameterisations, while we do our calculations explicitly. We did include an example case to demonstrate how especially the nitric acid partitions between different sized droplets in different parts of the cloud affects the composition of drizzle sized droplets within the cloud. Such modelling work has not been presented before. However, ensemble type simulations were left out for the follow-up paper with varying aerosol composition and cloud types with different amounts of surface precipitation to allow more robust comparison to existing wet scavenging parameterisations with the bulk approach. Such results provide more basis for further development of wet scavenging schemes employed on a global scale. For the same reason, we did not present here any results related to changes in cloud albedo due to the presence of semivolatiles. However, we will do this in the revised manuscript. We will present examples both on slightly precipitating stratocumulus deck and also cumulus type cloud field. We will analyze the results in more detail starting from the enhancement of droplet concentration, leading to changes in precipitation, cloud albedo and wet scavenging efficiencies of different compounds

**1.2 Specific Comments**

– Section 2.3.1 and 2.3.2 can go into an appendix as they are not original developments (Jacobson, 1997, 2002, 2005)

This we can do, and we will concentrate more on the novelty of the practical implementation of equations. We will also detail the unique use of the APD method for water, as explained above, in the revised manuscript.

– Section 3.1: Kokkola et al., 2018 have already developed confidence in the SALSA condensation routine. What is new here?

The difference here is that the partitioning of water is implemented in SALSA in a way that it is done using condensation equations with the LES time step of 1 second. SALSA versions in Kokkola et al. (2008, 2018) are employed in global models, they can be only used in subsaturated conditions and the partitioning of water is calculated solving the thermodynamical equilibrium of water between gas and particles, not condensation. In LES version of SALSA (see Tonttila et al. (2017)), we have extended SALSA to include also cloud and precipitating droplets and condensation equations are solved only for those hydrometeors. For aerosol, the same equilibrium assumption is used as in the global model configuration. There are two fundamental differences between Tonttila et al. (2017) and this version of SALSA in that the condensation was solved using 50 sub time step and in Tonttila et al. (2017) it used the APC scheme for the condensation of water and that scheme was not stable under high relative humidity. As explained above, APC has oscillatory behaviour especially in the water uptake on small aerosol particles. This was however improved by substituting the APC with the APD scheme. Although the APD scheme is not meant for water condensation, it, however, worked pretty well compared

to the APC in that regard. Due to the difficulty of solving condensation numerically efficiently while maintaining needed accuracy, we felt that the comparison against a detailed model needed to be presented.

In addition, none of the previous SALSA versions included the co-condensation of semi-volatile gases in aerosol, clouds and precipitation. For the simultaneous condensation of ammonia and nitric acid, condensation limiters were also implemented in order to get a stable numerical solution also for the smallest particle sizes. This will be emphasized in more detail

– P11L33: "we configured SALSA to nucleate cloud droplets"; isn't this the parameterizations of Abdul-Razzak and Ghan in SALSA?

This part will be rewritten to avoid misunderstanding. The "we configured SALSA" being mentioned here is not referring to some sort of novel configuration or scheme, but only to the switching-on of the cloud nucleation scheme which also includes the activation of a subpopulation of particles within a single size bin. To this end in Section 3.2, we only presented results for the updated condensation scheme to reproduce condensation on different sized particles. SALSA can employ the Abdul-Razzak and Ghan scheme, but as in large-eddy simulations we solve the condensation equation and thus resolve the supersaturation, we do not employ any parameterization but simply activate all particles that have grown larger than the critical size for activation in model resolved supersaturation.

– Definition of CEF: What are the assumptions and resultant errors associated with using a simple ratio of CDNsvc:CDN0 as a quantitative estimator of the effect on cloud formation? Or in other words, in Fig. 3(a), CEF = 1 falls within±of CEF w.r.t.[SVC]; is there then a (demonstrated) dependence?

This comment is not very clear to us, but we guess it may benefit from a comment we have given on the last bullet point of these responses about the CEFs behaviour. Also, the figure might be misleading and we will reproduce it using different statistical measures like percentiles.

– Section 3.3: The discussion here is merely descriptive and does not provide insight.

This section describes the evaluation of the model and because of that is mainly descriptive. In the revised version, we will elaborate that our scheme although using only 10 size bins to cover the whole size range from 3nm to approximately 10µm matches very well with the detailed parcel model.

– Section 4.2: Why, if these (P16L06: "1 ppb of HNO3 and 1 ppb of NH3") values of SVCs were relevant, were they not in the range of evaluation of SALSA-standalone?

The particle concentration selected for Figure 3a was too low that higher gas-phase concentration would not be realistic. However, we will adjust the particle phase concentrations and extend the range of concentration in the evaluation plots. We had conducted an extensive set but wanted to keep the evaluation section as short as possible in the manuscript.

– P16L12: "There is a notable increase in CDNC, which happens because the dissolved gases increase the sizes especially those of interstitial aerosols, which then reduces their corresponding critical saturation ratios (Kokkola et al., 2003b), and hence increasing the number of aerosols that can be activated into cloud droplets." What and how notable is the increase in CDNC? Where and by how much is the increase in aerosol sizes? Kokkola et al., 2003b is not needed here. What is the ostensible increase in CCN? What the reader would be interested in is a discussion supported by quantitative/statistical

values.

*Yes, the wording or description of these results will be improved in the revised paper; however, the figure in question here is Fig. 5, and it clearly shows by how much and where the respective parameters have changed between the two simulations. But like mentioned, we will clarify this description more in the revised version.*

– P16L19: "Of course, other conditions upon which the experiments were conducted are different but the results compare reasonably well." Yet again is a statement that appears ambiguous due to the absence of details/numbers

*We will remove this sentence about comparison as it can not be quantitatively done. Instead, we will compare against the droplet concentration estimated based on the detailed cloud parcel model and the probability density function for updrafts at the cloud base. Through this comparison, we can quantify if the average increase in CDNC from UCLALES-SALSA agrees with the cloud parcel model results, or does the cloud dynamics enhance of buffer the effect.*

– A general issue throughout the paper is that the write-up associated with the figures are descriptive in nature. While this is alright to guide the reader through the figures, the absence of a quantitative discussion (things that answer questions such as"How much more/less?", "How significant?", "Why/How much contribution of purported cause-effect?") makes the discussion appear speculative. Furthermore, a detailed discussion supported by statistics may uncover insights/interesting scientific points that are not currently demonstrated.

*This is an insightful and positive comment, it will guide us immensely in revising the manuscript.*

**1.3 Technical corrections**

– Table 1: Please recheck the values of the mean and standard deviation of particle diameters. There seems to be a copy-paste error. Additionally, standard deviation has the same units of measurement as the variable.

*Yes, there is such an error in the presented parameters. This will be corrected in the revised manuscript.*

– Figure 1: (a) x-axis: Confirm if radius or diameter. Also, the units reported seem wrong

*These are radii and yes, the units will be corrected.*

– Figure 4: There is a panel label for the figure which is unnecessary and presently mislabeled.

*The submitted manuscript included misplaced texts below figures and above figure captions. We will also revise all figures which had extra labelling in the manuscript*

– Figure 5: It is unclear how the ratios (right panels) can go below 0. Also, the units reported for precipitation rate is W/m2; is this the cooling due to precipitation or change in clear-sky radiative cooling or just an error and should be kg/s/m^2?

*Yes, the ratios need to be clarified further to express that its a fractional change, i.e (SV - CTRL)/CTRL. The precipitation flux can be presented also in the energy units from which the conversion to the mass unit can be made through latent*

heat. As we were mainly interested in the relative change, we did not perform this especially as the precipitation rate is very low. However, as suggested, it could be more easily understood if presented as kg/s/m^2

– Figure 7: Please recheck the legend of panel (b)

We will fix the figures in the revised manuscript.

– P01L15-24: References can be cleaned up: either they may be omitted if the above is cited, since all the information has moved from literature to textbooks, or the more pertinent and initial studies should be cited.

We will improve the use of references in the revised version.

– P01L17: IPCC AR5, 2013 would be a better "scientific" reference than Pörtner et al.,2019, which, although more recent, is a summary for policymakers.

This will be changed accordingly

– P02L21: Please add a line about changing atmospheric ammonia for completeness

This can be done, and in addition complemented with a discussion how it affects the aerosol acidity

– P03L13: Period after "Section" to be removed. Also "the last section of the paper" to be replaced with "Section 5"

We will correct this

– P14L09: Please remove the comma.

We will correct this

– P17L01-02: "Fig. 6 shows the domain averaged" is out of place.

We will fix the beginning of this paragraph

– P17L03: "depended" to "dependent"

We will correct this

– P17L03: "It is obvious that..." It may not be obvious to the reader. Please remove these words and then discuss why is it so.

We will elaborate that the partitioning of ammonia from gas-to-particle is very much driven by acidity and in this case, will condense to particles to neutralize sulfate. Instead, the partitioning of nitric acid depends very much on the water content of particles and its partitioning to particles increases with increasing humidity.

– P17L05: "It is apparent that..." It may not be apparent to the reader. Please remove these words and then discuss why is this so.

This will be corrected as explained in the reply to the previous referee comment.

**2 Referee 2**

**2.1 General Comments**

– The authors present a study in which they have developed a large eddy simulator to include the condensation of inorganic semi-volatile compounds (SVCs) condensing on a particle population to assess the impact it has on cloud droplet formation and subsequently cloud microphysics. Owing to the current lack of an appropriate data set,the LES predictions are evaluated (favorably) against those acquired from cloud parcel model simulations which allow for a higher particle size resolution.

The authors find the expected result that cloud droplet nucleating potential is increased by condensation of these SVCs due to increased particle size/soluble mass, and consequently a cloud droplet spectrum with reduced mean size and therefore precipitation. In this regard the paper does not present a serious lurch forward in understanding these processes/mechanisms. It would have been nice to see with the inclusion of condensing semi-volatile organic compounds in a VBS framework simultaneously, so as to ascertain whether or not there would be any interesting interaction effects, though most likely just a compounded effect on CCN activity.

At present I do not see that the paper is suitable for publication in ACP as it does not contain any new scientific insight. That said, it is nice to see these processes included in an LES and a more substantial study would certainly benefit from the increased process resolution.

The idea of adding a VBS framework simultaneously with inorganic semivolatiles is something we will work on. Detailed 3D atmospheric model and sectional aerosol will set some limits due to computational cost, but certainly, such studies are underway conducted. However, it has to be noted that solving condensation for different VBS compounds is trivial compared to calculating condensation of nitric acid and ammonia simultaneously.

We, however, disagree with the strong statement that paper does not contain any new scientific insight. We have a new technical implementation to solve difficult modelling problem (please see the details in the reply for Reviewer 1). In addition, as far as we know, nobody has presented how these semivolatile compounds partition between gas, aerosol, cloud and precipitating droplets by accounting for the size-dependent condensation dynamics. This affects how big a fraction of semivolatiles ends up in the precipitating droplets within the cloud. Although the current manuscript did not provide any new parameterizations, it brought this up and highlighted how the composition of drizzle droplets differs from bulk cloud water. To gain more insight ensemble type simulations are needed, which were originally planned as a follow-up for this paper presenting that we finally have a model that can be used for such a purpose.

**2.2 Specific Comments**

– Figure 2: Would be useful to clarify in the caption/y-axis that this is the change in bin radius (as opposed to number concentration)

We will clarify this in the revised manuscript

– Figure 5: mean droplet sizes less than 0?

No, the droplet sizes are not below zero. If you are referring to the last column of Fig 5, it shows the relative difference of the respective parameters, i.e. differences of simulations with SVs and the control simulation over the control simulation (SV - CTRL)/CTRL.

– Figure 7a: Could clarify in the caption that the parcel model curve correspond to "total" molar ratio

This will be clarified

– Figure 3: Would be interesting to see the total variability, i.e. min/max, as well. Also, might be worth noting in the text that the CEF is negative in some cases and elaborate why this might be.

We will make the plot again to avoid confusion. In the plot, we have presented the mean CEF (solid curve and the standard deviation being shown by the area shaded in pink). In practice, small negative values could be produced due to small oscillation in the condensation of different gases and activation scheme using only 10 sections to cover the whole size range. Using such an activation scheme requires interpolation of particle number and wet size distribution within a single bin, and that is difficult without including the detailed information of input size distribution. However, as presented in Figure 4, on average the effect of semivolatile is captured really well considering the low size resolution of SALSA.

Otherwise, we consider that the decrease in the number of droplets could be caused only by the presence of giant CCN, and condensation of HNO3 on those to effectively decrease supersaturation. We haven't included those in this test setup.

**References**

Jacobson, M. Z., Tabazadeh, A., and Turco, R. P.: Simulating equilibrium within aerosols and nonequilibrium between gases and aerosols, Journal of Geophysical Research: Atmospheres, 101, 9079–9091, 1996.

Kokkola, H., Korhonen, H., Lehtinen, K., Makkonen, R., Asmi, A., Järvenoja, S., Anttila, T., Partanen, A.-I., Kulmala, M., Järvinen, H., et al.: SALSA–a Sectional Aerosol module for Large Scale Applications, Atmospheric Chemistry and Physics, 8, 2469–2483, 2008.

Kokkola, H., Kühn, T., Laakso, A., Bergman, T., Lehtinen, K. E., Mielonen, T., Arola, A., Stadtler, S., Korhonen, H., Ferrachat, S., et al.: SALSA2. 0: The sectional aerosol module of the aerosol-chemistry-climate model ECHAM6. 3.0-HAM2. 3-MOZ1. 0, Geosci. Model Dev. Discuss, 2018.

Luo, G., Yu, F., and Schwab, J.: Revised treatment of wet scavenging processes dramatically improves GEOS-Chem 12.0. 0 simulations of surface nitric acid, nitrate, and ammonium over the United States, Geoscientific Model Development, 12, 3439–3447, 2019.

Luo, G., Yu, F., and Moch, J. M.: Further improvement of wet process treatments in GEOS-Chem v12. 6.0: impact on global distributions of aerosols and aerosol precursors, Geoscientific Model Development, 13, 2879–2903, 2020.

Tonttila, J., Maalick, Z., Raatikainen, T., Kokkola, H., Kühn, T., and Romakkaniemi, S.: UCLALES-SALSA v1. 0: a large-eddy model with interactive sectional microphysics for aerosol, clouds and precipitation, Geoscientific Model Development, 10, 169, 2017.

---

## Editor Comment (EC1) · Thorsten Bartels-Rausch (Editor) · 21 Jan 2021

Thanks for tackling and answering the referee comments and describing a way to make the manuscript stronger. I appreciate this. Because the 2 referee were very strong in their rejection, I would welcome a resubmission. I'm sorry to have to reject the manuscript.

Best regards Thorsten Bartels-Rausch
* * *
[Figure]

2020.